# Interactions between the Nociceptin and Toll-like Receptor Systems

**DOI:** 10.3390/cells11071085

**Published:** 2022-03-23

**Authors:** Lan Zhang, Ulrike M. Stamer, Melody Ying-Yu Huang, Frank Stüber

**Affiliations:** 1Department of Anaesthesiology and Pain Medicine, Inselspital, Bern University Hospital, University of Bern, 3010 Bern, Switzerland; ulrike.stamer@dbmr.unibe.ch (U.M.S.); ying-yu.hedinger@dbmr.unibe.ch (M.Y.-Y.H.); frank.stueber@insel.ch (F.S.); 2Department for BioMedical Research, University of Bern, 3008 Bern, Switzerland

**Keywords:** cell cultures, inflammation, nociceptin, nociceptin receptor, pain, toll-like receptors

## Abstract

Nociceptin and the nociceptin receptor (NOP) have been described as targets for treatment of pain and inflammation, whereas toll-like receptors (TLRs) play key roles in inflammation and impact opioid receptors and endogenous opioids expression. In this study, interactions between the nociceptin and TLR systems were investigated. Human THP-1 cells were cultured with or without phorbol myristate acetate (PMA 5 ng/mL), agonists specific for TLR2 (lipoteichoic acid, LTA 10 µg/mL), TLR4 (lipopolysaccharide, LPS 100 ng/mL), TLR7 (imiquimod, IMQ 10 µg/mL), TLR9 (oligonucleotide (ODN) 2216 1 µM), PMA+TLR agonists, or nociceptin (0.01–100 nM). Prepronociceptin (*ppNOC*), *NOP*, and *TLR* mRNAs were quantified by RT-qPCR. Proteins were measured using flow cytometry. PMA upregulated *ppNOC* mRNA, intracellular nociceptin, and cell membrane NOP proteins (all *p* < 0.05). LTA and LPS prevented PMA’s upregulating effects on *ppNOC* mRNA and nociceptin protein (both *p* < 0.05). IMQ and ODN 2216 attenuated PMA’s effects on *ppNOC* mRNA. PMA, LPS, IMQ, and ODN 2216 increased NOP protein levels (all *p* < 0.05). PMA+TLR agonists had no effects on NOP compared to PMA controls. Nociceptin dose-dependently suppressed TLR2, TLR4, TLR7, and TLR9 proteins (all *p* < 0.01). Antagonistic effects observed between the nociceptin and TLR systems suggest that the nociceptin system plays an anti-inflammatory role in monocytes under inflammatory conditions.

## 1. Introduction

Nociceptin and the nociceptin receptor (NOP) share high homology with opioid ligands and the classical opioid receptors, respectively, but have distinct profiles. Nociceptin has diverse effects in different species and pain models, depending on site of injection and dosage, with hyperalgesic effects at the supraspinal level and antinociceptive effects in the periphery [1,2,3,4]. An immunomodulatory role of nociceptin and NOP in circulating blood was described [1,5,6,7,8,9]. However, how the nociceptin system is regulated and which mechanisms are involved has not been fully elucidated.

Toll-like receptors (TLRs) are a family of pattern recognition receptors identified as important mediators during inflammation and pain processing [10,11]. TLR signaling provoked pain, and specific *TLR*-knockout mice showed attenuation of nociception compared to the wild type in various neuropathic pain models [11,12]. TLRs are abundantly expressed in human peripheral blood leukocytes and play a central role in immune response. Peripheral blood mononuclear cells (PBMC) of patients suffering from chronic pain demonstrated increased responsiveness to TLR ligands in vitro [13].

There is evidence that activation of µ-opioid receptors modulates TLR signaling, and opioids activate TLR2 or TLR4 signaling [14,15,16]. Furthermore, LPS stimulation increased the release of endogenous opioids from human monocytes in vitro [17]. Whereas many studies have focused on the crosstalk between the TLR system and opioids, no data on the interactions between nociceptin and TLRs are currently available.

Human monocyte-like THP-1 cells is a cell line derived from peripheral blood of a childhood case of acute monocytic leukemia. In our preliminary experiments, constitutive expression of nociceptin, the nociceptin receptor, and TLR (TLR2, TLR4, TLR7, and TLR9) proteins were determined in THP-1 cells. In addition, higher basal intracellular TLR7 and TLR9 protein levels were detected in THP-1 cells compared to human monocytic MM6 cells. We hypothesized that specific TLR signaling contributes to the regulation of nociceptin and the nociceptin receptor under inflammatory conditions, and that the nociceptin system has an influence on TLR expression. 

## 2. Materials and Methods

### 2.1. Cell Cultures

Human monocytic THP-1, U937, and promyelocytic HL-60 cells (all from CLS, Eppel-heim, Germany) were maintained in RPMI-1640 medium supplemented with 10% fetal bovine serum (FBS), 2 mM L-glutamine, 100 U/mL penicillin, and 100 µg/mL streptomycin at 37 °C and 5% CO_2_ atmosphere (Sigma-Aldrich, Buchs, Switzerland). For human monocytic MM6 cells (Thermo Scientific, Reinach, Switzerland), the same culture medium, plus 10 µg/mL insulin, 1 mM sodium pyruvate, and 1× non-essential amino acids, were used (Sigma-Aldrich, Buchs, Switzerland). Cells were seeded in flat-bottom 24-well plates at a density of 3 × 10^5^ cells per well in 1 mL of culture medium (TPP, Trasadingen, Switzerland).

#### 2.1.1. Cell Line Screening and Dose-Response Experiments

In our previous studies, phorbol myristate acetate (PMA), a potent proinflammatory activator, was the only immune activator with an upregulating effect on nociceptin in human monocytic MM6 cells and peripheral blood leukocytes [18,19]. To examine the impact of PMA on mRNA expression of nociceptin and *NOP* in different cell lines, MM6, THP-1, U937, and HL-60 cells were screened. Flow cytometry analysis showed that both nociceptin and NOP proteins could be detected in each cell line. To investigate the impact of PMA on nociceptin and NOP expression in these cell lines, cells were cultured with or without PMA 5 ng/mL for 24 h (Sigma-Aldrich, Buchs, Switzerland), and mRNA expression of nociceptin precursor (prepronociceptin, *ppNOC*) and *NOP* were detected. The PMA concentration used for screening was based on our previous studies [18,19]. 

As a result of the screening experiment, THP-1 cells were chosen and then cultured with or without various concentrations of PMA (0.01–100 ng/mL) for 24 h, and *ppNOC* and *NOP* mRNA levels were quantified. Based on these dose-response experiments, PMA 5 ng/mL was used in the subsequent experiments.

#### 2.1.2. Cell Viability

THP-1 cells were cultured with or without 0.01–1000 ng/mL PMA, 100 ng/mL of lipopolysaccharide (LPS) (Sigma-Aldrich, Buchs, Switzerland), 10 µg/mL of lipoteichoic acid (LTA), 10 µg/mL of imiquimod (IMQ), 1 μM of oligonucleotide (ODN) 2216 (InvivoGen, Toulouse, France), or 0.01–100 nM of nociceptin (Phoenix Pharmaceuticals, Karlsruhe, Germany) for 24 h. Viable cells were determined by a CytoFLEX S Flow Cytometer (Beckman Coulter, Nyon, Switzerland) using fixable viability dye eFluor™ 506 according to the manufacturer’s instructions (eBioscience, Vienna, Austria), and cell viability was presented as a percentage of the untreated samples.

#### 2.1.3. Co-Stimulation of THP-1 with PMA and TLR Agonists

To examine the contribution of TLR signaling to the regulation of nociceptin and NOP under inflammatory conditions, cells were cultured with or without 10 μg/mL of LTA, 100 ng/mL of LPS, 10 μg/mL of IMQ, or 1 μM of ODN 2216, with or without 5 ng/mL of PMA, for 24 h. The concentrations of the TLR agonists were based on previous studies [18,19,20,21] and the current cell viability assay. Samples were collected after 24 h, and cells were prepared for RNA isolation or quantification of protein levels by flow cytometry.

#### 2.1.4. Stimulation of THP-1 with Nociceptin

To investigate the effects of nociceptin on TLRs, cells were cultured with or without 0.01–100 nM of exogenous nociceptin. After 24 h, cell surface TLR2 and TLR4 proteins, as well as intracellular TLR7 and TLR9 proteins, were measured.

### 2.2. RNA Isolation, cDNA Synthesis and Relative Quantification

Total RNA was isolated using a high pure RNA isolation kit following the manufacturer’s protocol (Roche, Rotkreuz, Switzerland). RNA concentrations and purity were measured using a NanoDrop 2000 (Thermo Scientific, Reinach, Switzerland). Subsequently, cDNA was synthesized (Transcriptor High Fidelity cDNA Synthesis Kit, Roche, Rotkreuz, Switzerland), and *ppNOC*, *NOP*, and *TLR* mRNAs were detected (Table 1).

Reference genes hypoxanthine phosphoribosyl-transferase 1 (*HPRT1*) and glyceraldehyde-3-phosphate dehydrogenase (*GAPDH*) were selected as internal controls. RT-qPCR reactions were performed in duplicate in 384-well plates by a LightCycler^®^ 480 (Roche, Rotkreuz, Switzerland) using 5 µL of 2× LightCycler^®^ 480 Probes MasterMix and 0.5 µL RealTime ready Assay in a final volume of 10 µL. cDNA prepared from human SK-N-DZ cells served as a calibrator.

Standard curves were generated separately for each target gene and reference gene, using serial dilutions of cDNA template known to express the gene of interest in high abundance. mRNA levels were analyzed using the advanced relative quantification module of the LightCycler^®^ 480 software (Version 1.5, Roche, Rotkreuz, Switzerland). Target gene mRNA levels were computed based on the qPCR amplification efficiency and the crossing point difference, and calculated as the ratio of the target gene/*HPRT1* and *GAPDH* of each sample normalized to the calibrator used in the PCR reactions (normalized ratio). 

### 2.3. Flow Cytometry

#### 2.3.1. Measurement of Cell Membrane Proteins

Cells were collected by centrifugation (300× *g*, 5 min) at 4 °C and washed with ice cold PBS. To reduce the possible non-specific binding background, cells were suspended with 10% of human serum and incubated for 20 min on ice (Sigma-Aldrich, Buchs, Switzerland).

For NOP staining, 10 μL of cell suspension was transferred to a 96-well U-bottom plate (TPP, Trasadingen, Switzerland) and treated with anti-NOP mAb (Sigma-Aldrich, Buchs, Switzerland) or isotype-control antibody (BD Biosciences, Allschwil, Switzerland) at a final concentration of 5 µg/mL with 50 µL hypotonic saponin solution (50 µg/mL of saponin, 130 mM of sucrose, 50 mM of KCL, 50 mM of sodium acetate, 20 mM of HEPES with DI water at 3:2 ratios, all from Sigma-Aldrich) for 5 min on ice [22]. After washing with permeabilization buffer (PBS with 1% FBS, 1% saponin, 1% sodium azide, all from Sigma-Aldrich), cells were incubated with anti-NOP mAb or isotype-control antibody (BD Biosciences, Allschwil, Switzerland) at a final concentration of 5 µg/mL for 1 h on ice. Subsequently, cells were washed three times with permeabilization buffer and stained with 1 μg/mL of PE-conjugated secondary antibody (Thermo Scientific, Reinach, Switzerland) for 1 h on ice in the dark. After the staining, cells were washed and fixed with 1% paraformaldehyde (PFA, Sigma-Aldrich, Buchs, Switzerland). 

To stain cell surface TLR, cells were incubated with PE-labelled anti-TLR2, anti-TLR4 mAb, or isotype-control antibody (BD Biosciences, Allschwil, Switzerland) at a final concentration of 1 µg/mL for 1 h on ice in the dark, washed and fixed in 1% PFA.

#### 2.3.2. Measurement of Intracellular Proteins

Intracellular staining of nociceptin was performed as previously described [18,19]. Briefly, cells were fixed with 1.5% PFA, permeabilized (BD Cytofix/Cytoperm™ Kit, BD Biosciences, Allschwil, Switzerland), and stained with anti-nociceptin antibody (Phoenix Pharmaceuticals, Karlsruhe, Germany) or isotype-control antibody (Abcam, Cambridge, UK) at a final concentration of 5 μg/mL for 1 h at room temperature (RT). Samples were washed three times and stained with 1 μg/mL of PE-conjugated secondary antibody (BD Biosciences, Allschwil, Switzerland) for 1 h at RT in the dark. Cells were then washed and suspended in staining buffer.

As for intracellular TLR7 and TLR9, cells were prepared using the BD Cytofix/Cytoperm^TM^ Kit according to the protocol, and stained with PE-conjugated anti-TLR7, anti-TLR9 mAb, or the respective isotype-control antibodies at a final concentration of 1 µg/mL for 1 h at RT in the dark (Thermo Scientific, Reinach, Switzerland). Subsequently, they were washed and suspended in staining buffer.

Cells with the same preparation but without any staining were used as negative controls to determine the autofluorescence. All flow cytometric measurements were performed using a CytoFLEX S Flow Cytometer (Beckman Coulter Life Sciences, Krefeld, Germany). A total of 10,000 cells in the gated population were recorded per sample. Mean fluorescence intensity, which represents the expression level of the target proteins, was calculated using the FlowJo V10 software (TreeStar Inc., Ashland, OR, USA).

### 2.4. Statistical Analysis

Statistical analysis was performed using STATISTICA 10.0 (StatSoft, Inc., Tulsa, OK, USA). Data are presented as box-and-whisker plots showing medians, interquartile range (IQR), 10–90 percentiles, and mean. Kruskal-Wallis with post hoc test (Dunn’s) and Wilcoxon test with correction for multiple testing where it applies. *p* < 0.05 was considered statistically significant.

## 3. Results

### 3.1. Nociceptin and NOP Expression in Different Cell Lines

The basal levels of *ppNOC* mRNA in MM6, THP-1, U937, and HL-60 were below the detection limit, whereas intracellular nociceptin proteins could be detected. NOP was constitutively expressed in these cell lines at mRNA and protein levels (Figure 1A,B). PMA at 5 ng/mL significantly upregulated *ppNOC* mRNA in MM6, THP-1, and HL-60 cells after 24 h, compared to untreated controls. The highest upregulating effect of PMA on *ppNOC* mRNA was observed in THP-1 cells (Figure 2A). As for NOP, PMA upregulated its mRNA expression in MM6 and U937 cells after 24 h, compared to controls (Figure 2B). According to these preliminary results, THP-1 was chosen for use in the subsequent experiments.

### 3.2. Dose-Dependent Effects of PMA

PMA dose-dependently upregulated *ppNOC* mRNA in THP-1 cells after 24 h. Compared to untreated controls, *ppNOC* mRNA levels were upregulated either by PMA 1 ng/mL or PMA 10 ng/mL. Maximum *ppNOC* mRNA expression was seen in THP-1 cells stimulated with 10 ng/mL PMA (Figure 3A). As for NOP, a slight upregulation of *NOP* mRNA expression was observed only in the samples treated with PMA 1 ng/mL for 24 h (Figure 3B). Based on these results, PMA 5 ng/mL was used in the subsequent experiments.

### 3.3. Cell Viability

Flow cytometry analysis demonstrated that PMA (0.01–1000 ng/mL) dose-dependently decreased viable cell count in THP-1 cells after 24 h (Figure 4A). LTA at 10 µg/mL, LPS at 100 ng/mL, IMQ at 10 µg/mL, ODN 2216 at 10 µM, PMA at 5 ng/mL, or 0.01–100 nM of nociceptin had no influence on THP-1 cell viability, compared to untreated controls. Viability of the THP-1 cells exposed to LTA, LPS, IMQ, ODN 2216, or different concentrations of nociceptin up to 24 h was >93%. In addition, cell viability in 5 ng/mL of PMA treated samples exceeded 89% (Figure 4B).

### 3.4. Nociceptin, NOP, and TLR Expression in THP-1

*NOP*, *TLR2*, *TLR4*, *TLR7*, and *TLR9* mRNAs were constitutively expressed in THP-1 cells, whereas *ppNOC* mRNA was below the level of detection (Figure 5A). Cell membrane NOP, TLR2, and TLR4, as well as intracellular nociceptin, TLR7, and TLR9 proteins could be detected by flow cytometry (Figure 5B,C).

### 3.5. Effects of TLR Agonists on the Nociceptin System

To examine effects of TLR signaling on nociceptin and NOP expression, TLR agonists specific for TLR2 (LTA), TLR4 (LPS), TLR7 (IMQ), or TLR9 (ODN 2216) were employed. RT-qPCR data showed that none of these TLR agonists had an impact on *ppNOC* mRNA level. However, an increase of intracellular nociceptin was measured in all TLR agonist groups, compared to controls (all *p* < 0.05). PMA upregulated *ppNOC* mRNA (Figure 6) as well as intracellular nociceptin protein levels (Figure 7A,B) after 24 h, compared to controls (both *p* < 0.0001). The TLR2 agonist, LTA, and the TLR4 agonist, LPS, completely abolished PMA’s upregulating effects on *ppNOC* mRNA and intracellular nociceptin proteins, compared to the samples treated with PMA only (all *p* < 0.05). In addition, *ppNOC* mRNA expression in PMA+IMQ and PMA+ODN 2216 were decreased to 24.2 (20.3–29.1)% and 82.9 (70.1–95.8)%, compared to the PMA group (both *p* < 0.05) (Figure 6 and Table 2). In contrast, IMQ and ODN 2216 had no antagonistic effects on intracellular nociceptin upregulation by PMA (Figure 7B).

TLR agonists, PMA, and PMA+TLR agonists had no impact on *NOP* mRNA (Table 2). An increase in cell membrane NOP proteins was detected in the cells stimulated with LPS, IMQ, ODN 2216, or PMA, compared to controls (Figure 7C). No changes of cell membrane NOP proteins were observed in the samples co-stimulated with PMA+TLR agonists, compared to the PMA group (Figure 7C).

### 3.6. Effects of Activation of the Nociceptin System on TLR Expression

To investigate the contribution of the nociceptin system to TLRs, cells were cultured with or without different concentrations of nociceptin (0.01–100 nM) for 24 h. Flow cytometry analysis revealed that nociceptin dose-dependently suppressed TLR2, TLR4, TLR7, and TLR9 proteins. Cell surface TLR2 was attenuated by the highest concentration of nociceptin, compared to controls (*p* < 0.05 Figure 8A). Suppression of TLR4 was observed in the cells cultured with nociceptin within a larger range of concentrations (0.1–100 nM) (Figure 8B). Nociceptin at amounts of 1–100 nM downregulated intracellular TLR7 and TLR9 proteins (Figure 8C,D).

## 4. Discussion

This study focuses on the interactions between the nociceptin and TLR systems in human monocytic THP-1 cells. The results show that TLR signaling prevents the upregulation of nociceptin by PMA, and nociceptin suppresses TLR protein expression.

Nociceptin and the nociceptin receptor proteins are constitutively expressed in THP-1 cells and regulated either by PMA or TLR agonists in the present study. Specific TLR agonists prevented PMA’s upregulating effects on *ppNOC* mRNA and intracellular nociceptin protein. Nociceptin dose-dependently decreased cell surface TLR2 and TLR4 as well as intracellular TLR7 and TLR9 proteins. These results support previous findings that nociceptin and NOP are regulated in blood leukocytes under inflammatory conditions and play a regulatory role during immune response [6,7,8,19,23,24]. 

Published evidence suggests that nociceptin plays an immune regulatory role. However, detailed information on mediator/receptor systems involved is lacking. Clinical data indicated that nociceptin can be detected in human synovial fluid and plasma, with lower levels in the synovial fluid. However, only extracellular nociceptin levels were measured [25]. Another study showed that nociceptin mRNA was expressed in human peripheral mononuclear neutrophils (PMN), with nociceptin evoking PMN chemotaxis and recruitment [26]. The effects of nociceptin on monocytes/macrophages still have to be elucidated. In our previous study, regulation of nociceptin and the nociceptin receptor by inflammatory mediators (LPS, cytokines) in human peripheral blood cells was observed [8]. In addition, PMA significantly upregulated nociceptin at mRNA and protein levels in human monocytic MM6 cells as well as in peripheral blood leukocytes [18,19]. Moreover, solid evidence from preclinical and clinical studies confirms that compounds targeting the nociceptin system are effective therapeutic approaches for substance abuse and potential candidates for pain management [27,28,29,30,31]. Previous studies mainly focused on the role of nociceptin and NOP in neural tissues; much less is known about their functions in blood immune cells. To the best of our knowledge, no data on the interactions between TLRs and the nociceptin system in a monocytic cell line have been published up to now.

### 4.1. PMA-Induced THP-1 Model

THP-1 cells have been widely used to study immune response and signaling pathways, and activated THP-1 cells provide an alternative to peripheral blood monocyte models [32]. TPH-1 cells were chosen in the present study because constitutive expression of nociceptin, NOP, and TLR proteins could be detected with higher basal TLR7 and TLR9 protein levels, compared to MM6 cells. Moreover, a more pronounced upregulation of *ppNOC* mRNA by PMA was observed in THP-1 cells, compared to MM6 cells.

In contrast to cell lines, ex vivo whole blood cells can demonstrate cross-talk between different blood cells, can interact with blood components, and may only represent a single blood donor, which may lead to misinterpretation of results. Therefore, PMA-induced THP-1 cells seem to be a suitable model to study the effects of the nociceptin system on TLRs, and conversely the effects of TLR’s on nociceptin and the nociceptin receptor.

### 4.2. Effects of TLR Signaling on the Nociceptin System

Clinical data have shown aberrant expression of nociceptin and NOP in blood of patients suffering from pain and inflammatory disease [1,5,6,7,23,24]. However, mechanisms underlying their regulation still need to be identified.

The cross-talk between opioids and TLRs has been discussed previously [11,14,33,34]. Suppression of *TLR4* mRNA by morphine in mouse RAW cells and peritoneal macrophages were reported [35]. In another study, TLR4 signaling acted as a transient counter-regulator for inflammatory pain in vivo and increased the release of endogenous opioids from human monocytes in vitro [17]. In addition, there was evidence that TLR-antagonistic drugs may attenuate opioid-induced side effects [36].

The present data demonstrate that the basal level of *ppNOC* mRNA in THP-1 cells was below the detection limit, whereas intracellular nociceptin protein could be detected. These results are in line with previous findings, indicating low nociceptin mRNA expression in resting human peripheral blood neutrophils and storage of preformed nociceptin protein in the cells [37].

In the current model, PMA significantly upregulated nociceptin, both at mRNA and protein levels. Activation of TLR2 or TLR4 signaling completely blocked the PMA-mediated increase in *ppNOC* mRNA as well as intracellular nociceptin protein levels. In addition, TLR7 and TLR9 agonists partially prevented PMA’s upregulating effects on *ppNOC* mRNA expression.

Interestingly, in a rat model of neuropathic pain, TLR2 and TLR4 antagonists produced analgesia and improved the analgesic effects of buprenorphine [33]. In addition to the well-known dual interaction with mu and kappa opioid receptors [38,39], buprenorphine also seems to be a partial agonist for the nociceptin receptor and antagonist for the delta opioid receptor [4,29,40,41,42]. This suggests that TLRs may indeed play a role in the regulation of endogenous nociceptin during immune response in vivo.

In the present study, the upregulation of nociceptin by PMA was more pronounced for mRNA than for intracellular proteins. Extracellular secretion of nociceptin may be the reason for this difference, as nociceptin is secreted by cells under inflammatory conditions [18,19,37].

Although TLR agonists had no effects on *ppNOC* mRNA expression, an increase in intracellular nociceptin protein was detected in all TLR agonist-treated samples. One possible explanation might be poor correlation of nociceptin mRNA and protein levels. In addition, proteolytic processing of the nociceptin precursor protein might be enhanced under inflammatory conditions.

In contrast, neither PMA nor TLR agonists affected *NOP* mRNA expression in THP-1 cells. However, cell membrane NOP protein levels were upregulated in the cells stimulated with these mediators. This is consistent with the results from a previous study which found that LPS/PepG decreased *NOP* mRNA but increased NOP protein in human umbilical vein endothelial cells [43]. As intracellular nociceptin was upregulated and secreted by THP-1 cells after PMA stimulation, the released nociceptin may bind to NOP and participate in autoregulation of the cells [44,45].

### 4.3. Nociceptin Effects on TLRs

There is growing evidence of the involvement of nociceptin and NOP in pain and sepsis [1,5,6]. Whereas the regulation of the nociceptin system and related mechanisms have been well characterized, information on the interactions between the nociceptin system and TLRs is still lacking.

The inhibitory effects of nociceptin on TLRs suggest that the nociceptin system may play an anti-inflammatory role in blood during immune response. In a rat model of colitis, peripheral injection of low-dose nociceptin had protective effects, while higher dose nociceptin worsened colitis [46]. In a mouse model of inflammatory bowel disease, oral administration of a NOP agonist showed anti-inflammatory and antinociceptive effects [47]. In contrast, inhibition of NOP decreased the severity of symptoms in a mouse colitis model [48], and systemic administration of nociceptin increased mortality in a rat sepsis model [49]. Furthermore, increased plasma nociceptin concentrations in septic patients and higher nociceptin levels in non-survivors have been reported [23,24]. In a previous study, increased *NOP* and decreased *ppNOC* mRNAs were detected in peripheral blood leukocytes from end-stage cancer patients and septic patients [7]. However, TLR expression levels were not assessed in these studies.

TLRs were regulated in blood leukocytes of patients suffering from pain or infectious diseases [50,51]. Increased responsiveness of PBMCs to in vitro TLR2, TLR4, and TLR7 activation has been reported in chronic pain patients [52]. Moreover, mortality in sepsis was associated with downregulation of TLR2 levels in blood monocytes [53]. Thus, increased TLR activation may affect the upregulation of nociceptin in blood cells under inflammatory conditions, and conversely, the increased nociceptin may contribute to the downregulation of TLR expression.

The present study has some limitations. First, only intracellular nociceptin proteins were measured. Extracellular nociceptin proteins will be addressed in a future project, as nociceptin may be secreted by the cells after stimulation. Second, the translational value of the present cell culture model needs to be considered. Although THP-1 cells are a suitable alternative to monocytes, the cultures may not accurately reflect the modulation of nociceptin and *NOP* in blood cells under pathophysiological conditions in vivo. Yet, this cell line provides a stable in vitro model [54], which enables the study of mechanisms of nociceptin and *TLR* regulation. Nevertheless, the reciprocal negative regulation observed between the nociceptin and TLR systems in THP-1 cells emphasizes the translational potential of new therapeutic targets in the treatment of pain and/or inflammation. Future studies should investigate interactions between these two systems in blood leukocyte subsets.

## 5. Conclusions

The present investigation highlights antagonistic effects observed in the nociceptin and TLR systems, suggesting that the nociceptin system may play an anti-inflammatory role in leukocytes during the immune response. Fundamental insights into the crosstalk between the nociceptin system and TLRs may shed new light on the treatment of pain and/or inflammatory disease.

## Figures and Tables

**Figure 1 cells-11-01085-f001:**
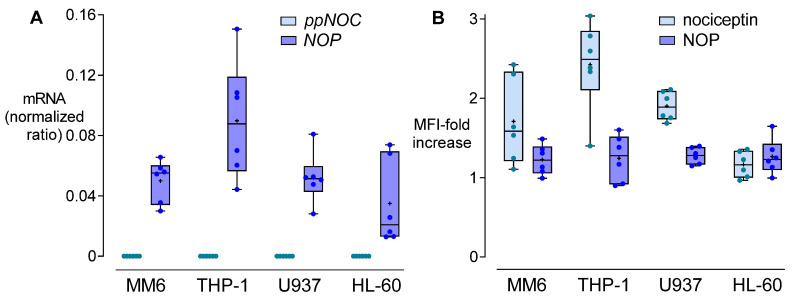
Constitutive expression of nociceptin and NOP in different human cell lines. (**A**) Prepronociceptin (*ppNOC*) and *NOP* mRNA expression in human monocytic cell line MM6, THP-1, U937, and lymphoblast cell line HL60. (**B**) Intracellular nociceptin and cell membrane NOP protein levels in four cell lines. Flow cytometry data are presented as mean fluorescence intensity (MFI) related to the respective isotype controls. Boxplots with individual data points, median, IQR, and mean “+”; *n* = 6.

**Figure 2 cells-11-01085-f002:**
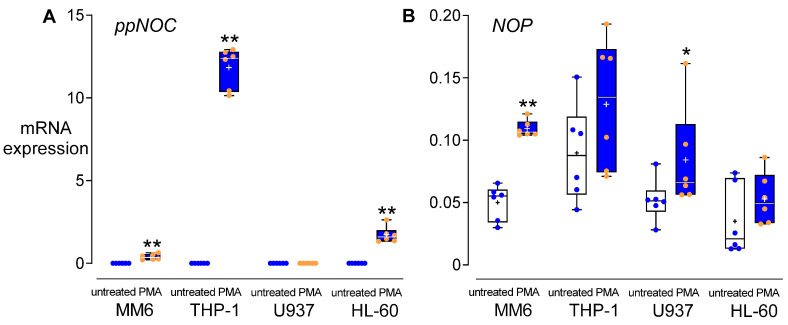
PMA effects on *ppNOC* and *NOP* mRNA expression in different human cell lines. Cells were treated with 5 ng/mL of PMA or without PMA (controls) for 24 h. *ppNOC* mRNA expression (**A**) and *NOP* mRNA expression (**B**) in different cell lines with or without PMA. Boxplots with individual data points, median, IQR, and mean “+”; *n* = 6. Wilcoxon test, *, *p* < 0.05; **, *p* < 0.01, compared to the respective controls.

**Figure 3 cells-11-01085-f003:**
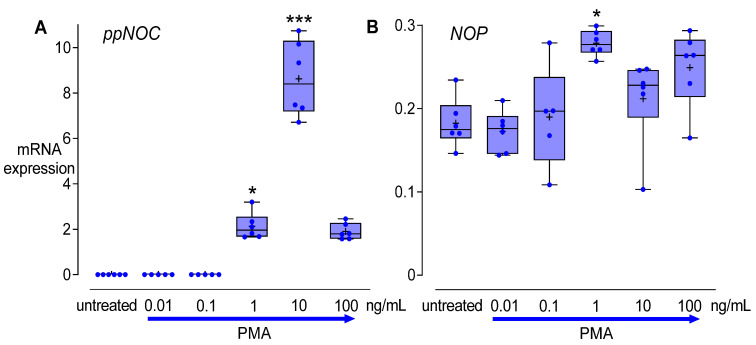
PMA dose-dependent effects on *ppNOC* and *NOP* mRNA expression. THP-1 cells were cultured with different concentrations of PMA (0.01–100 ng/mL) or without PMA (control) for 24 h. mRNA levels of *ppNOC* (**A**) and *NOP* (**B**) were determined. Boxplots with individual data points, median, IQR, and mean “+”; *n* = 6. Kruskal-Wallis test with post hoc test (Dunn’s). *, *p* < 0.05, ***, *p* < 0.001, compared to the respective controls (untreated).

**Figure 4 cells-11-01085-f004:**
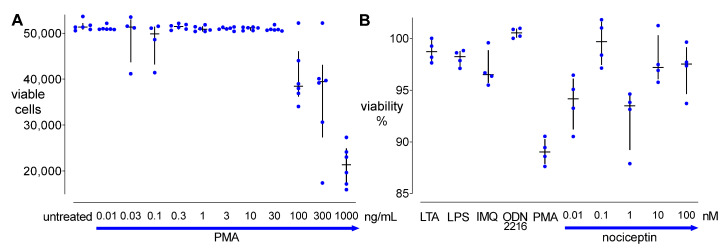
Cell viability of THP-1 cells exposed to different stimuli. (**A**) Viable cells in the samples treated with or without PMA (0.01–1000 ng/mL) for 24 h, *n* = 6. (**B**) Cell viability was measured in THP-1 cells following 24 h of incubation with or without different stimuli (LTA 10 µg/mL, LPS 100 ng/mL, IMQ 10 µg/mL, ODN 2216 1 µM, and nociceptin 0.01–100 nM), *n* = 4. Flow cytometry data are presented as percentage of viable cells related to the untreated group (control). Dots, median, and IQR.

**Figure 5 cells-11-01085-f005:**
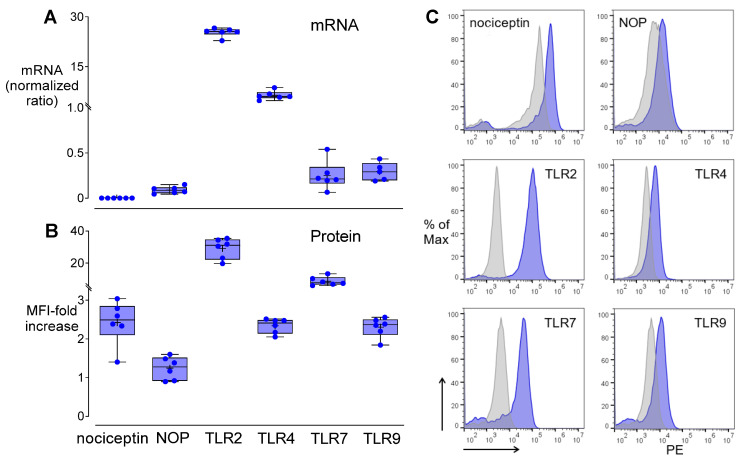
Constitutive nociceptin, NOP, TLR expression patterns in THP-1 cells. (**A**) Basal levels of *ppNOC*, *NOP*, *TLR2*, *TLR4*, *TLR7*, and *TLR9* mRNA expression. (**B**) Basal levels of nociceptin, NOP, and TLR proteins. (**C**) Representative histogram plots of cell membrane NOP, TLR2, TLR4, and intracellular nociceptin, and TLR7 and TLR9 protein levels. Cells were stained with specific antibodies for nociceptin, NOP, TLR2, TLR4, TLR7, or TLR9 (blue-filled histogram). The grey-filled histogram represents samples stained with respective isotype control antibodies. Flow cytometry data are presented as mean fluorescence intensity (MFI) related to the respective isotype controls. Boxplots with individual data points, median, IQR, and mean “+”; *n* = 6.

**Figure 6 cells-11-01085-f006:**
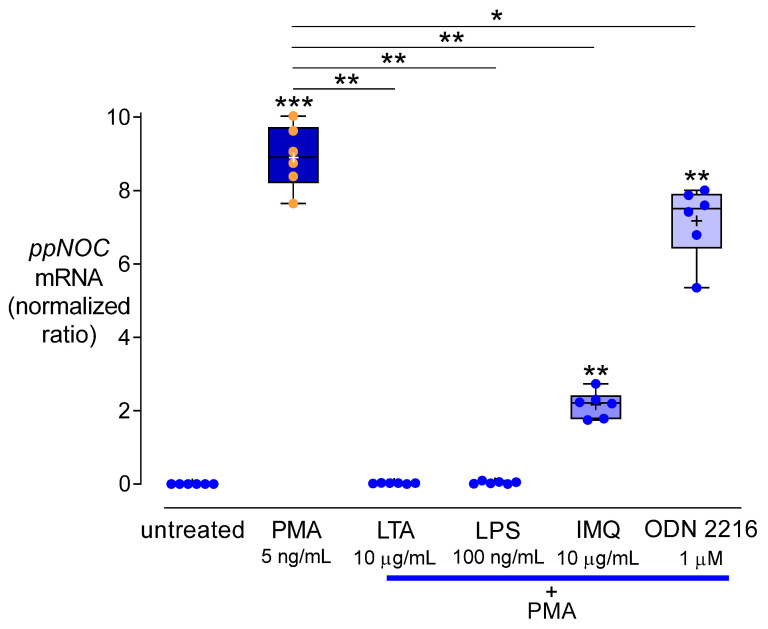
Effects of TLR agonists on the regulation of *ppNOC* mRNA by PMA in THP-1 cells. Cells were cultured with or without PMA and with or without different TLR agonists specific for TLR2 (LTA), TLR4 (LPS), TLR7 (IMQ), and TLR9 (ODN 2216) for 24 h. Boxplots with individual data points, median, IQR, and mean “+”; *n* = 6. Wilcoxon test with correction for multiple testing. *, *p* < 0.05; **, *p* < 0.01; ***, and *p* < 0.001.

**Figure 7 cells-11-01085-f007:**
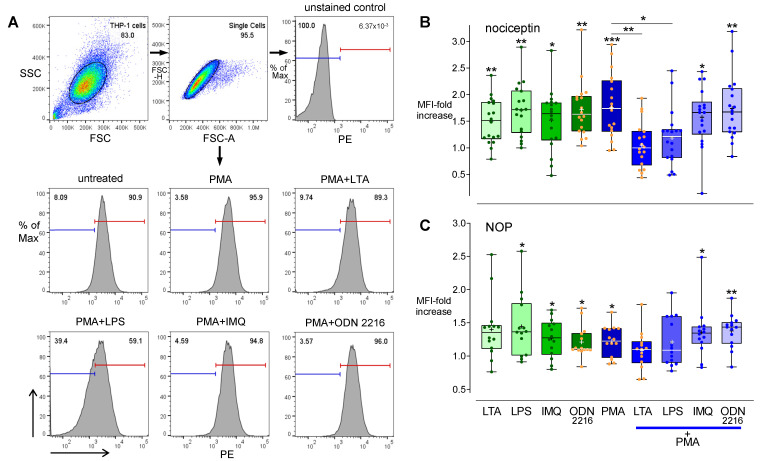
Effects of TLR agonists on the regulation of nociceptin and NOP proteins by PMA in THP-1 cells. Cells were cultured with or without different TLR agonists specific for TLR2 (LTA 10 µg/mL), TLR4 (LPS 100 ng/mL), TLR7 (IMQ 10 µg/mL), TLR9 (ODN 2216 1 µM), and with or without PMA for 24 h. (**A**) Representative histogram plots of intracellular nociceptin protein levels in THP-1 cells treated with or without PMA 5 ng/mL and PMA+TLR agonists (blue and red lines: setting of gates). Flow cytometric analysis of intracellular nociceptin (**B**) and membrane NOP protein levels (**C**). Flow cytometry data are presented as mean fluorescence intensity (MFI) related to the respective untreated groups. Boxplots with individual data points, median, IQR, and mean “+”; *n* = 18. Wilcoxon test with correction for multiple testing. *, *p* < 0.05; **, *p* < 0.01; ***, *p* < 0.001.

**Figure 8 cells-11-01085-f008:**
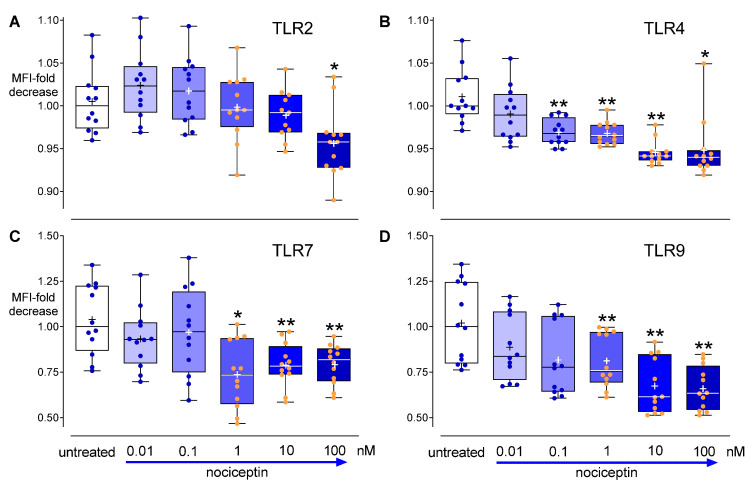
Effects of nociceptin on TLR protein levels in THP-1 cells. Cells were cultured with or without different concentrations of exogenous nociceptin (0.01–100 nM) for 24 h. Flow cytometric analysis of cell surface TLR2 (**A**), TLR4 (**B**), and intracellular TLR7 (**C**) and TLR9 protein levels (**D**). Flow cytometry data are presented as mean fluorescence intensity (MFI) related to the corresponding untreated groups (controls). Boxplots with individual data points, median, IQR, and mean “+”; *n* = 12. Wilcoxon test with correction for multiple testing. *, *p* < 0.05; **, *p* < 0.01, compared to the respective controls.

**Table 1 cells-11-01085-t001:** Sequence of primers and RT-qPCR reaction conditions.

Gene		Primer Sequence 5′ to 3′	RT-qPCR Reaction Conditions
*HPRT1*	Fwd	TGACCTTGATTTATTTTGCATACC	
	Rev	CGAGCAAGACGTTCAGTCCT	
*GAPDH*	Fwd	AGCCACATCGCTCAGACAC	
	Rev	GCCCAATACGACCAAATCC	
*ppNOC*	Fwd	GGACAGCTTCGACCTGGAG	pre-incubation
	Rev	TGACCTTGGTGCATGGAGT	95 °C, 10 min
*NOP*	Fwd	CCCAAGGAGGTTGCAGAA	amplification (45 cycles)
	Rev	GCCGTAGATAACCTCCCAGA	95 °C, 10 s
*TLR2*	Fwd	CTCTCGGTGTCGGAATGTC	60 °C, 30 s
	Rev	AGGATCAGCAGGAACAGAGC	72 °C, 1 s
*TLR4*	Fwd	CAAGATGCCCCTTCCATTT	
	Rev	TCCTTAGGAATTAGCCACTAGACTTT	
*TLR7*	Fwd	GCCCCCAAGATGGTTTAAG	
	Rev	GCATCCCCAATTTCTTTGG	
*TLR9*	Fwd	CGCTACTGGTGCTATCCAGA	
	Rev	AGCCCAGGGAGGAGCTAAG	

*HPRT1*, hypoxanthine phosphoribosyl-transferase 1; *GAPDH*, glyceraldehyde-3-phosphate dehydrogenase; *ppNOC*, prepronociceptin; *TLR*, toll-like receptor.

**Table 2 cells-11-01085-t002:** Effects of TLR agonists on the regulation of *ppNOC* and *NOP* mRNA expression by PMA in THP-1 cells.

Stimulation	*ppNOC*	*NOP*
untreated	0	0.08 (0.07/0.10)
LTA 10 µg/mL	0	0.05 (0.04/0.08)
LPS 100 ng/mL	0	0.04 (0.04/0.06)
IMQ 10 µg/mL	0	0.09 (0.07/0.11)
ODN 2216 1 µM	0	0.05 (0.03/0.07)
PMA 5 ng/mL	**8.90 (8.20/9.73)**	0.04 (0.03/0.06)
PMA+LTA	0.03 (0.01/0.03) **	0.04 (0.04/0.05)
PMA+LPS	0.04 (0.01/0.07) **	0.04 (0.03/0.06)
PMA+IMQ	**2.21 (1.77/2.41)** **	0.05 (0.04/0.08)
PMA+ODN 2216	**7.51 (6.43/7.90)** *	0.06 (0.02/0.11)

Data are presented as median with IQR, *n* = 6. *ppNOC* mRNA levels significantly increased compared to the untreated group (printed in bold). Wilcoxon test with correction for multiple testing. *, *p* < 0.05; **, *p* < 0.005, compared to the samples treated with PMA only. IMQ, imiquimod; LPS, lipopolysaccharide; LTA, lipoteichoic acid; *NOP*, nociceptin receptor; OND 2216, oligonucleotide 2216; PMA, phorbol myristate acetate; *ppNOC*, prepronociceptin.

## Data Availability

The data in support of the findings of this study are available from the corresponding author upon reasonable request.

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
