# Peer review of "Interactions between the Nociceptin and Toll-like Receptor Systems"

_cells, 2022, doi:10.3390/cells11071085_

Round 1

Reviewer 1 Report

Dear authors,

Thank you for answering my questions. In the revised manuscript, second paragraph of Introduction (information about PMA) can be moved to the last paragraph of Introduction where PMA is first mentioned. 

Author Response

Response to Reviewers    cells-1622875

Reviewer 1

Comments and Suggestions for Authors

Dear authors,

Thank you for answering my questions. In the revised manuscript, second paragraph of Introduction (information about PMA) can be moved to the last paragraph of Introduction where PMA is first mentioned. 

Thank you for the comments. We now felt it most appropriate to move the extended paragraph on PMA to the “Materials and Methods” section.

All information about PMA is now provided, lines 66-79.

“In our previous studies, phorbol myristate acetate (PMA), a potent proinflammatory activator, was the only immune activator with an upregulating effect on nociceptin in human monocytic MM6 cells and peripheral blood leukocytes [18,19]. To examine the impact of PMA on mRNA expression of nociceptin and NOP in different cell lines, MM6, THP-1, U937, and HL-60 cells were screened. Flow cytometry analysis showed that both nociceptin and NOP proteins could be detected in each cell line. To investigate the impact of PMA on nociceptin and NOP expression in these cell lines, cells were cultured with or without PMA 5 ng/ml for 24 hours (Sigma-Aldrich, Buchs, Switzerland), and mRNA expression of nociceptin precursor (prepronociceptin, ppNOC) and NOP were detected. The PMA concentration used for screening was based on our previous studies [18,19].

As a result of the screening experiment, THP-1 cells were chosen and then cultured with or without various concentrations of PMA (0.01-100 ng/ml) for 24 hours, and ppNOC and NOP mRNA levels were quantified. Based on these dose-response experiments, PMA 5 ng/ml was used in the subsequent experiments.”

According to the results of these pilot experiments, THP-1 cells were chosen for subsequent experiments.

Reviewer 2 Report

This manuscript by Zhang et al. aims to determine the interactions between the nociceptin and toll-like receptor (TLR) system. The authors used a human monocyte cell line, THP-1, to determine the impact of nociceptin or TLR agonists on the expression of receptors. They found that PMA+TLR agonists had no effects on NOP compared to PMA controls. Nociceptin dose-dependently suppressed TLRa, TLR4, TLR7, and TLR9 proteins. However, there are several major concerns.

  1. This study only determine the impact of the nociceptin or TLR agonists on the expression of the receptors. However, the expression of TLR may not directly reflect to the function of those receptors. This study didn't provide any data related to the impact of those agonists on the function.
  2. Treatment with PMA directs the THP-1 to macrophage-type cells, and it expects that the expression of TLRs will be increased in the PMA-treated THP-1.
  3. This study lacks of mechanistic data about how the nociception and TLRs interact with each other.
  4. The authors should provide the cell viability data.
  5. Provide a rationale for choosing 5 ng/mL PMA.
  6. The paper need to be edited by native English speaker with science background.

Author Response

Response to Reviewers    cells-1622875

Reviewer 2

Comments and Suggestions for Authors

This manuscript by Zhang et al. aims to determine the interactions between the nociceptin and toll-like receptor (TLR) system. The authors used a human monocyte cell line, THP-1, to determine the impact of nociceptin or TLR agonists on the expression of receptors. They found that PMA+TLR agonists had no effects on NOP compared to PMA controls. Nociceptin dose-dependently suppressed TLRa, TLR4, TLR7, and TLR9 proteins. However, there are several major concerns.

  1. This study only determined the impact of the nociceptin or TLR agonists on the expression of the receptors. However, the expression of TLR may not directly reflect to the function of those receptors. This study didn't provide any data related to the impact of those agonists on the function.

Thank you for the comment. Indeed, in this study, as a first step, we focused on whether TLR expression is impacted by nociceptin. And yes, it is very interesting to know whether nociceptin also has an impact on the function of toll-like receptors. As a main result, we found substantial downregulation of TLRs by nociceptin, which may lead to an overall quantitative reduction of functional TLRs and, therefore, may reduce the organism’s response to TLR activators. We are very interested in addressing this point in a forthcoming study.

  1. Treatment with PMA directs the THP-1 to macrophage-type cells, and it expects that the expression of TLRs will be increased in the PMA-treated THP-1.

THP-1 cells constitutively express TLRs on their surface, as do primary monocytes and many monocytic cell lines. PMA activates monocytes to differentiate in part to macrophages and even increases their TLR expression so that cells will probably react to inflammatory stimuli with more sensitivity. This is what we detect in THP-1 cells stimulated by PMA: TLR expression increases (data not shown). Yet the primary endpoint in these PMA-stimulated cells is the upregulation of nociceptin.  This is quite unique and not detected when studying “classical” TLR activators such as LPS (TLR4), LTA (TLR2), imiquimod (TLR7) or oligonucleotides 2216 (TLR9). Therefore, THP-1 cells are suitable to study regulation of nociceptin expression. Studying the impact of nociceptin on TLR expression, PMA stimulation was not used. So, here we show the impact of nociceptin on constitutive TLR expression.

  1. This study lacks of mechanistic data about how the nociception and TLRs interact with each other.

In the present investigation, we aimed to examine whether TLR signaling contributes to the regulation of nociceptin and the nociceptin receptor in human monocytic THP-1 cells, and to investigate possible effects of nociceptin which is induced in immune cells under inflammatory conditions. This first approach reveals a downregulation of TLRs by nociceptin. In a forthcoming study, we aim to investigate whether this effect is caused directly by activated NOP, which may also suppress TLR-activating factors. Thus, we are very interested in the mechanisms involved. Also, mechanisms of TLR signaling—which impacts the nociceptin system and conversely the influence of nociceptin on TLRs—will be addressed in oncoming projects designed to unravel a possible “negative feedback loop” in the inflammatory network cascades.

  1. The authors should provide the cell viability data.

Cell viability results are presented in the “Results”, lines 207-213.

“Flow cytometry analysis demonstrated that PMA (0.01-1000 ng/ml) dose-dependently decreased viable cell count in THP-1 cells after 24 hours (Figure 4A). LTA 10 µg/ml, LPS 100 ng/ml, IMQ 10 µg/ml, ODN 2216 10 µM, PMA 5 ng/ml, or nociceptin 0.01-100 nM had no influence on THP-1 cell viability compared to untreated controls. Viability of the THP-1 cells exposed to LTA, LPS, IMQ, ODN 2216, or different concentrations of nociceptin up to 24 hours was >93%. In addition, cell viability in PMA-treated samples exceeded 89% (Figure 4B).”

An additional figure is added showing cell viability in THP-1 cells cultured with different concentrations of PMA and cell viability of THP-1 cells treated with different stimuli (Figure 4).

  1. Provide a rationale for choosing 5 ng/mL PMA.

Results of the current PMA dose-response experiment, lines 194-197.

“PMA dose-dependently upregulated ppNOC mRNA in THP-1 cells after 24 hours. Compared to untreated controls, ppNOC mRNA levels were significantly upregulated by PMA 1-10 ng/ml. Maximum ppNOC mRNA expression was seen in cells stimulated with 10 ng/ml PMA (Figure 3A).”

Data of the cell viability experiment, lines 207-208 and 212-213.

“Flow cytometry analysis demonstrated that PMA (0.01-1000 ng/ml) dose-dependently decreased viable cell count in THP-1 cells after 24 hours (Figure 4A).”

“In addition, cell viability in PMA 5 ng/ml treated samples exceeded 89% (Figure 4B).”

Based on these results, PMA 5 ng/ml was used in the subsequent experiments.

  1. The paper need to be edited by native English speaker with science background.

The manuscript has now undergone editing by a native English speaker with 20 years of experience in the editing of scientific papers. We now find the language has improved considerably.

Reviewer 3 Report

It is interesting to see the interactions between the nociceptin and TLR systems in human monocytic THP-1 cells. The manuscript clearly suggests that TLR signalling prevents the upregulation of nociceptin by PMA, and nociceptin suppresses TLR expression. The same team has previously reported that inflammatory mediators influence the expression of nociceptin and its receptor in human whole blood cultures (PLoS One. 2013 Sep 16;8(9):e74138.). Several other groups have also established the potential role for nociceptin in modulating inflammation. (J Orthop Surg Res. 2020 Jul 16;15(1):266.  J Immunol. 2001 Mar 15;166(6):3650-4.). Authors may need to clearly discuss the differentiations from the previous studies. In addition, the study design is not strong and there are several issues mentioned below that need to be addressed.

  1. The method mentions that “Cells were seeded at a density of 3×105 cells/ml in flat-bottom 24-well plates”. Is this statement correct?

  1. Authors report that “The basal levels of ppNOC mRNA in MM6, THP-1, U937 and HL-60 were below the detection limit, whereas intracellular nociception proteins could be detected.” Did the authors try to check the protein expression by immunofluorescence staining? It will be interesting to have that data set here.

  1. The study is based on the observance of expression of ppNOC mRNA. Authors need to present the expression at the level of protein. Also, each expression data needs to be presented as bars with individual data points.

  1. Why did the authors use nan-parametric tests for statistical analysis, especially in Figures 2, 3, and 5? Authors need to clearly describe (with justification) the statistics used for each panel.

  1. As presented in Fig. 3, PMA dose-dependently upregulated ppNOC mRNA in THP-1 cells after 24 hours. It will be interesting to see effects of doses on viability. Also, the expression again needs to be checked at the level of protein. Again, the authors fail to convince with the statistics shown for this figure.

6. Cell viability needs also to be tested by staining or enzymatic way.

Author Response

Response to Reviewers    cells-1622875

Reviewer 3

Comments and Suggestions for Authors

It is interesting to see the interactions between the nociceptin and TLR systems in human monocytic THP-1 cells. The manuscript clearly suggests that TLR signalling prevents the upregulation of nociceptin by PMA, and nociceptin suppresses TLR expression. The same team has previously reported that inflammatory mediators influence the expression of nociceptin and its receptor in human whole blood cultures (PLoS One. 2013 Sep 16;8(9):e74138.). Several other groups have also established the potential role for nociceptin in modulating inflammation. (J Orthop Surg Res. 2020 Jul 16;15(1):266. J Immunol. 2001 Mar 15;166(6):3650-4.). Authors may need to clearly discuss the differentiations from the previous studies. In addition, the study design is not strong and there are several issues mentioned below that need to be addressed. 

Thank you for the references.

J Orthop Surg Res. 2020;15(1):266.

Nociceptin levels were measured in the synovial fluid and plasma of 20 patients undergoing total knee arthroplasty. Results showed that nociceptin proteins can be detected in synovial fluid and plasma, with a significant lower concentration in synovial fluid compared to plasma samples.

J Immunol. 2001;166(6):3650-4.

Nociceptin mRNA was detected in human peripheral blood polymorphonuclear leukocytes (PMN) and monocytes. Nociceptin effects on PMN chemotaxis and recruitments were observed.

PLoS One 2013;8(9):e74138.

Regulation of nociceptin and the nociceptin receptor by different inflammatory mediators (LPS, cytokines) were investigated. We found that LPS as well as cytokines suppress mainly NOP and, in part, PNoc mRNA expression in human whole blood cultures.

A paragraph has been added in the “Discussion”, lines 298-308 and 312-314.

“Published evidence suggests that nociceptin plays an immune regulatory role. However, detailed information on mediator/receptor systems involved is lacking. Clinical data indicated that nociceptin can be detected in human synovial fluid and plasma, with lower levels in the synovial fluid. However, only extracellular nociceptin levels were measured [25]. Another study showed that nociceptin mRNA was expressed in human peripheral mononuclear neutrophils (PMN) with nociceptin evoking PMN chemotaxis and recruitment [26]. The effects of nociceptin on monocytes/macrophages still have to be elucidated. In our previous study, regulation of nociceptin and the nociceptin receptor by inflammatory mediators (LPS, cytokines) in human peripheral blood cells was observed [8]. In addition, PMA significantly upregulated nociceptin at mRNA and protein levels in human monocytic MM6 cells as well as in peripheral blood leukocytes [18,19].”

…. “To the best of our knowledge, no data on the interactions between TLRs and the nociceptin system in a monocytic cell line have been published up to now.”

  1. The method mentions that “Cells were seeded at a density of 3×105 cells/ml in flat-bottom 24-well plates”. Is this statement correct?

The sentence has been modified, lines 63-64.

“Cells were seeded in flat-bottom 24-well plates at a density of 3×105 cells per well in 1 ml of culture medium.”

  1. Authors report that “The basal levels of ppNOC mRNA in MM6, THP-1, U937 and HL-60 were below the detection limit, whereas intracellular nociception proteins could be detected.” Did the authors try to check the protein expression by immunofluorescence staining? It will be interesting to have that data set here.

In the present study, intracellular nociceptin was stained and detected using flow cytometry. We will try immunofluorescence staining of nociceptin in future studies to further address the location of intracellular nociceptin.

  1. The study is based on the observance of expression of ppNOC mRNA. Authors need to present the expression at the level of protein. Also, each expression data needs to be presented as bars with individual data points.

Nociceptin expression was detected at mRNA and protein levels in the present investigation. Data of intracellular nociceptin protein levels are presented in Figure 1, Figure 5 and Figure 7.

Thank you for the suggestion. Figures have been modified, expression data are presented as bars or box plots with individual data points.

  1. Why did the authors use nan-parametric tests for statistical analysis, especially in Figures 2, 3, and 5? Authors need to clearly describe (with justification) the statistics used for each panel.

Non-parametric analysis is a more conservative approach for experiments with low sample numbers, where a normal distribution of data is uncertain.

In case of small data sets, a test of significance for normality may lack power to detect the deviation of the variable from normality. Normality tests such as the Kolmogorov-Smirnov test will only have power when the number of data is more than 30. In this study, sample size was small (less than 20). Therefore, we prefer to use non-parametric tests for the data analysis.

  1. As presented in Fig. 3, PMA dose-dependently upregulated ppNOC mRNA in THP-1 cells after 24 hours. It will be interesting to see effects of doses on viability.

An additional figure (Figure 4) is added showing PMA effects on cell viability. 

In our setting at hand, regulation of nociceptin expression by different doses of PMA was detected at the mRNA level. We felt that nociceptin mRNA levels would reveal the dose-dependent response of THP-1 cells to PMA very sensitively in contrast to intracellular protein levels, which do not adjust as quickly and may underlie more complex processes of intracellular transportation, storage or excretion. We intend to study this dose dependency of intracellular protein levels in a follow-up project.

  1. Cell viability needs also to be tested by staining or enzymatic way.

Cell viability of THP-1 cells was determined by staining using flow cytometry in the present study. Data are presented on page 7, line 207-213 and Figure 4.

Round 2

Reviewer 2 Report

The paper has been improved, however, some major concerns brought up from the initial review are still not addressed.

Reviewer 3 Report

The authors have satisfactorily addressed most of my concerns.

This manuscript is a resubmission of an earlier submission. The following is a list of the peer review reports and author responses from that submission.

Round 1

Reviewer 1 Report

This is an interesting paper in the field of pain research. The subject is of high relevance and the design of the experiments and main conclusions are solid. The manuscript could be improved by a better explanation of the doses of the drugs elected in the experiments since a single dose was used. Furthermore since the experiments were performed in cells the translational value of the experiments should be critically considered. Finally, the references are not very recent and there are important papers in this field that could be added. For example, Ubaldi M et al., 2021 (Int. J. Mol. SCi), Tabata H 2021 (BMC Urol).

Reviewer 2 Report

Interactions between the nociception and toll-like receptor systems

Zhang et al. studied possible interactions between nociception and TLR systems using THP-1 cell line. Authors investigated whether TLR system is involved, at least in part, on the analgesic and anti-inflammatory as well as nociceptive effects of NOP/nociception system. They measured mRNA and protein levels of NOP, nociception, TLR2, TLR4, TLR7, TLR9 in cells treated with or without PMA using agonists for both systems. They found that both systems have inhibitory effect on each.

For a full consideration of publishing this manuscript, here are my suggestions and questions:

  1. For reader not familiar with the compounds and procedure (even though it was published previously), please explain what phorbol-myristate-acetate is and why it is used.

  1. What does THP-1 stand for? Please give some information about the cell line to the readers.

  1. Line 301: please correct as buprenorphine, dual mu opioid receptor agonist/kappa opioid receptor antagonist. Buprenorphine is not nociception/mu opioid receptor agonist.

  1. Please explain why THP-1, human leukemia monocytic cell line, was chosen over MM6 or human peripheral blood leukocytes in this study. Authors have shown previously measurement of NOP/nociception in these cells (references 16 and 17). Would not be better to use human peripheral blood leukocyte cell line to measure both NOP/nociception and TLR systems? It would be good to explain why they preferred THP-1 cell line for readers who do not study cell lines.

  1. Cell viability results were not shown or mentioned in the results section.

Reviewer 3 Report

This manuscript by Zhang et. al. aims to determine the interaction between nociceptin and toll-like receptors in THP-1 cells. The THP-1 cells were treated PMA, specific agonists for different TLRs, or nociceptin, and then the expression of prepronociceptin (ppNOC), NOP, and TLR mRNA and/or protein was measured. The authors determined some potential interactions between the two systems. However, there are several major issues.

1) Extensive studies on the interaction between nociceptin and TLRs were reported previously, and this study just repeated those studies in a different cell line without any mechanistic investigation.

2) The TLRs are continuously expressed in the cells, therefore, measurement of mRNA may not be appropriate. Determining the impact of nociceptin on the activation of those receptors should be considered.

3) Treatment with PMA causes the differentiation of THP-1 into macrophages, and the rationale for choosing the PMA-induced THP-1 model is not sufficient for this study.